**Data Availability Statement:** The data underlying the results presented in the study are available

# Effects of environmental awareness training and environmental commitment on firm's green innovation performance: Empirical insights from medical equipment suppliers

Qiong Wu[1], Senlin Xie[2], Shihan Wang[3], Anning Zhou[3], Lucille Aba Abruquah[4,5], Zhen Chen🔳[6]*

1 School of Management Science and Engineering, Chongqing Technology and Business University, Chongqing, China, 2 School of International Business, Chongqing Technology and Business University, Chongqing, China, 3 School of Business Administration, Chongqing Technology and Business University, Chongqing, China, 4 CERATH Development Organization, Accra, Ghana, 5 Research Global, Accra, Ghana, 6 Dazhou Central Hospital, Dazhou, China

* zhenchen2017@outlook.com

## Abstract

In recent years, the intensification of global industrialization coupled with the enterprise's production and operating activities have caused pollution, increasing the current environmental pressure. Relevant government departments in China have instituted several stringent measures (environmental protection sensitization and awareness activities, training sessions, and exchange activities targeted towards enterprise managers) to address these rising environmental problems. Though these measures have gained traction over the years, there is a dearth of research on their effectiveness on the green innovation performance of enterprises. To bridge the gap, this research explores the effect of environmental awareness training, knowledge exchange activities, and commitment on green innovation performance with survey data from 285 medical equipment manufacturing companies in China. It further expands the theoretical application of environmental awareness training, commitment, and innovation performance from the lens of the Knowledge-Based View. The findings depict a positive relationship between environmental awareness training and innovation performance. It also finds a mediating influence of environmental commitment in the relationship between environmental awareness training and green innovation performance. Furthermore, environmental knowledge exchange activities positively moderate the relationship between environmental awareness training and environmental commitment. These findings offer valuable insights for the green development of medical equipment manufacturing enterprises and the government to formulate environmental protection policies.

from (https://figshare.com/s/5ed7ec4ed8edeab92ad4)

**Funding:** This work was supported by the National Natural Science Foundation of Chongqing [Grant Number CSTB2022NSCQ-MSX0502]; and the City-School Cooperation Project of Dazhou Science and Technology Bureau [Grant Number DZKJJ2020S05]. The entire process and conclusions of this study were not influenced by the fund sponsor. The funders had no role in study design, data collection and analysis, decision to publish, or preparation of the manuscript.

**Competing interests:** The authors have declared that no competing interests exist.

## 1. Research background

The significance of green innovation cannot be overemphasized in developed and emerging economies. It plays a critical function in the twenty-first century for enterprises' success, development, and survival. Mo, Boadu [1] assert that green innovation manages the environment, energy usage, pollutant production, and waste disposal and recycling towards cleaner production and sustainability. Thus, it serves as a superior player, mitigates the production of pollutants, and increases cost efficiency and competitive advantage. For the past few years, stakeholders have gradually become aware of green innovation's importance to society and how enterprises should operate in an environmentally friendly manner to integrate green practices into strategic and process management. However, compared with developed markets environment, emerging markets face challenges between development and the environment. For instance, with the escalation of global industrialization, Chinese enterprises' production and operating activities have caused pollution, threatening the life of the earth's ecosystem [2]. Due to the alarming nature of the menace, for enterprises to survive, scholars, government agencies, and civil society organizations have recognized environmental protection sensitization and awareness training programs as a vital weapon for curbing the rising environmental issues to promote sustainability. Environmental awareness training refers to the sum of the training and learning actions on ecological issues, hands-on activities for sustainability, and relevant environmental policies and regulations taken by enterprises to meet environmental challenges. Environmental awareness training activities offer corrective measures to mitigate ecological menace towards innovation performance. Previous studies have established the positive impact of pro-environmental education on environmental protection [3, 4]. However, there is a dearth of studies on the environmental awareness training effectiveness on the green innovation performance (GIP) of enterprises, especially in emerging economies. This research aims to bridge the gap by exploring the cause-and-effect relationship between environmental awareness training (EAT), environmental knowledge exchange activities (EKEA), and environmental commitment (EC) in GIP and related boundary conditions using the Chinese medical equipment manufacturing industry. Chinese market environment offers a fertile ground for the study of such nature to understand the linkage amongst the variables.

First, research shows that education and training experiences are crucial in business management decisions [5, 6]. For instance, extant works have established the positive impact of pro-environmental education on environmental protection [3, 4]. However, there is still a lack of relevant evidence on the specific extent to which environmental training impacts GIP. Hence, the current study probes the linkage between EAT and GIP.

Second, this work contends that EC mediates between EAT and GIP. From the extant literature, research on EC primarily centers on the basic level of individuals and the environment. For example, Hojnik, Ruzzier [7] explore the determinants of individuals' inclination to pay for green energy. Cicatiello, Ercolano [8] explore the relationship between personal views on the environment and the environment of a specific region. Few studies have, however, linked EC with GIP to examine the impact of EC on EAT and GIP. This research thus investigates the involvement and efficacy of EC in the green innovation process of medical equipment enterprises to bridge the research gap.

Third, leveraging the knowledge-based view, this research probes into the expansion effect of EKEA on the relationship between EAT and EC. EKEA can aid enterprises in overcoming knowledge limitations that arise within a single entity [9], promote collaboration [10] and multilevel knowledge sharing [11], and effectively improve the green innovation performance of enterprises [12]. Hence, this scholarship examines the moderating effect of EKEA in the

model and expands the application of a knowledge-based view in environmental protection-related fields.

In summary, to solve the above problems, this scholarship uses survey data from 285 medical equipment manufacturing companies in China to investigate the mechanism of EAT's influence on GIP, the mediating role of EC and the positive regulatory impact of EKEA, and the logical relationship between these variables from the perspective of knowledge-based theory. The findings are expected to expand the applications of EKEA and EC in the green innovation field, making it an essential contribution to the theoretical and practical aspects of the subject. Thus, it provides the academic community, enterprises, and governments with a further understanding of the role of EAT, EC, and EKEA in green innovation planning. It also provides a decision-making reference for the green development of medical equipment manufacturing enterprises.

## 2. Literature review

### 2.1 Knowledge-based view

We draw from the knowledge-based view (KBV) to explore the relationship between environmental awareness activities and GIP. The KBV of the enterprise builds and extends the resource-based view (RBV) that emphasizes the strategic significance of knowledge as a unique resource [13]. The theory concentrates on how enterprises create, acquire, protect, transfer, and use knowledge [14] for higher performance. Thus, it is regarded as the convergence of the enterprise knowledge research flow. Proponents of KBV perceive knowledge as the most significant strategic organizational resource in terms of value creation. Although, lack of consensus over the nature of knowledge has prevented the KBV from developing into an integrated, stand-alone theory, but has not constrained its ability to provide penetrating insights into organizational strategy and management. Boadu, Xie [15] assert that the fundamental purpose of an organization is to create and apply knowledge that can offer the foundation for sustainable differentiation (i.e., difficult to imitate) to attain a competitive edge. The theory asserts that knowledge base and ability are the main determinants for enterprises to obtain results in the knowledge economy [16]. Though multifaceted, scholars have established that the amalgamation and the configuration of intangible resources, such as knowledge resources, are crucial to innovation [4]. Admittedly, KBV postulates that focusing on the efficiency of knowledge exchange and searching for the best method can boost the management and operation of business activities, leading to organizational innovation and competitive edge [14, 17]. Hence, the KBV perspective is suitable for scrutinizing environmental awareness knowledge exchange and GIP. It explicitly addresses how EAT affects the knowledge base of managers, which is reflected in the management's future business decisions [18], resulting in an enhanced EC and ultimately improving the GIP of enterprises.

To this end, with the KBV as a foundation, we have developed a research model on EKEA, EAT, EC, and GIP, as shown in Fig 1.

### 2.2 Environmental awareness training and green innovation performance

Green innovation performance measures the extent to which organizations develop innovations that condense or abate environmental damages, impact, and worsening while optimizing the use of natural resources [1]. Thus, it denotes the innovative practices of businesses in products, processes, management, and reduction of pollution emissions. As a strategy, it offers great opportunities to meet buyers' requirements while preserving the ecosystem [1]. Admittedly, the concept offers organizations a roadmap for achieving sustainable competitive advantages in an ecologically effective way [1, 19]. From the extant studies, green innovation has

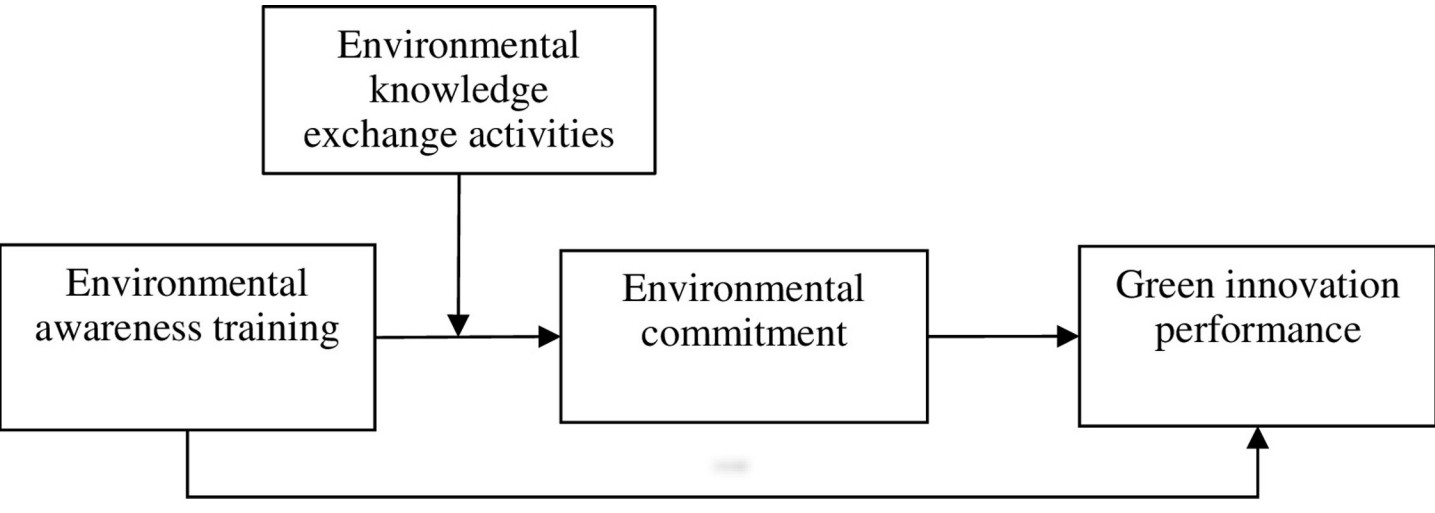

**Fig 1. Research framework.**

been categorized into two aspects, namely, product and process innovation [20]. However, the ultimate goal is to condense environmental risks in product and service functions for customers, resource utilization and cost efficiency, and organizational flexibility [21].

From the extant works, there is no clear definition of EAT. This paper draws from similar research on environment protection and training [22, 23] to define EAT as the sum of the training and learning actions on environmental issues, hands-on activities for sustainability, and relevant environmental policies and regulations taken by enterprises to meet environmental challenges. Drawing on the KBV, EAT can influence the manager's knowledge and environmental awareness by facilitating the exchange and learning of relevant knowledge, leading to improved green innovation performance [24, 25]. It is predominantly manifest in two main ways: First, it allows managers to acquire knowledge associated with preserving the environment (e.g., environmental concepts, culture, and values). Thus, EAT is more inclined towards green innovation-related strategies and actions in management decisions, which can promote the green innovation performance of enterprises [26, 27]. Second, there is evidence to suggest that organizational identity has a positive impact on GIP [28]. Previous studies have established the positive impact of pro-environmental education on environmental protection [4].

In this context, we contend that training on environmental protection in the Chinese medical equipment manufacturing sector can promote managers' willingness to participate in environmental protection activities, which, in turn, enhances the green corporate image and identity of enterprises and ultimately promotes GIP [29]. For this, we suggest the following hypothesis:

H1: Environmental awareness training has a positive impact on green innovation performance.

### 2.3 EAT and EC

Environmental commitment has gradually become one of the important subjects in environmental management [1]. Studies show that higher environmental education can stimulate citizenry environmental awareness [30] and promote environmental protection behavior [31, 32].

Drawing on the KBV, EAT can increase awareness and change in behavior, thus promoting EC [33]. For instance, environmental education organized by the government and society can

make managers realize the importance of environmental protection, thus boosting EC. Besides, environmental protection education provided by the government and society can strengthen the managers' sense of social responsibility, which, in turn, promotes EC. Previous studies have proved that regular knowledge acquisition activities and initiatives aimed at environmental issues positively influence environmental commitment [34].

Building on the KBV, we contend that the environmental awareness training process of the Chinese medical equipment manufacturing sector can boost environmental commitment. Hence, we postulate that EAT can affect EC in an organization. We suggest the following assumptions:

H2: Environmental awareness training has a positive impact on environmental commitment

## 2.4 EC and GIP

Linked to the above Hypothesis 2, drawing upon the KBV, environmental commitment is one of the most significant motives for individual intent to acquire knowledge about ecological deeds. EC refers to people's willingness to engage in environmental protection efforts. According to the KBV, EC can influence the enterprise's environmental management and green practices, which, in turn, leads to enhanced green innovation performance. It manifests in two ways: First, it can cultivate a sense of sustainable development within enterprises. This helps enterprises to create sustainable practices and invest necessary funds in developing green products, which ultimately promotes GIP [35, 36]. Second, consumers not only have a growing concern for the environment but also exhibit an increasing interest in purchasing green products [37]. EC can, thus, improve the organizational image of enterprises, competitiveness, and market share, which, ultimately promotes GIP. For instance, the findings of studies conducted by Mo, Boadu [1] and Lin and Ho [38] on managerial environmental concerns and green innovation performance and green innovation reveal significant impacts among the variables.

Drawing on the KBV, we contend that EC can aid enterprises in formulating wide-ranging ecological protection strategies in the direction of environmental matters, which, in turn, enhances enterprises' GIP. We therefore propose the following assumptions based on these assertions:

H3: Environmental commitment positively affects green innovation performance

## 2.5 Mediating role of EC in EAT and GIP

Studies have indicated that EAT is a continuous education that influences managers' environmental awareness and decision-making, ultimately shaping their environmental attitudes [3, 39]. Concerning external knowledge acquisition, EAT can help managers acquire knowledge related to environmental protection concepts and practices. Therefore, EAT can change their behavioral awareness [32], which affects their values and environmental attitudes [40], thus promoting EC.

EAT further establishes the value orientation of green development through EC, which enhances green product development of enterprises [41, 42] and ultimately promotes GIP. The current scholarship contends that EC can positively influence enterprises' green innovation activities. Thus, enterprises can adopt some responsible production techniques to enhance GIP. Extant works have proved how managerial environmental concerns affect the enterprise's GIP. For instance, Mo, Boadu [1] and Xie, Chen [4] find a snooping consequence of managerial environmental concerns in the correlation between pro-environmental education and eco-friendly agricultural production and CSR activities and GIP, respectively.

Drawing from the internal knowledge transfer, we contend that EAT can promote EC while in the internal knowledge sharing and exchange, it makes environmental protection the consensus of managers to establish the image of enterprises in the hearts of the people and the government, thus promoting GIP. We therefore suggest the hypothesis below:

H4: Environmental commitment mediates the relationship between environmental awareness training and green innovation performance

## 2.6 Moderating role of EKEA

EAT ensures the transfer of knowledge to enterprise managers through training. Training creates environmental protection awareness, which enhances the sense of social responsibility in enterprise managers. The enhanced social responsibility of enterprise managers improves their inclination to formulate ecological strategies, which ultimately promotes EC [43]. However, most knowledge transmitted to managers through education and training is explicit. Explicit knowledge is open and lacks creativity [44], which is not conducive to transmission and reception. On the contrary, tacit knowledge is the source of several opportunities and potentials found and created. Therefore, in knowledge management, it is a great challenge to make explicit knowledge implicit so that knowledge can be effectively received and used in the transmission process [45]. This paper introduces the variable EKEA, which can effectively solve this obstacle.

With reference to Buder [46], this paper defines EKEA as the behavior of changing one's environmental awareness or interacting with others. EKEA is a dynamic and flowing process that combines different forms of knowledge from multiple sources to effectively improve the efficiency and practicality of knowledge transfer [47]. Existing research has shown that EKEA is critical in extending the knowledge spillover effect within organizations and building competitive advantages [48]. Additionally, EKEA has the potential to facilitate collaboration between individuals with diverse backgrounds and skills, which results in the creation of new knowledge and more efficient sharing and utilization of existing knowledge. This enhances the effectiveness of knowledge transfer and acquisition [49]. Therefore, when EKEA is limited, the knowledge acquired by managers is mainly reflected at the explicit level, which results in low efficiency of knowledge transmission and acceptance. This adversely affects the impact of EAT on EC. Conversely, when there is an abundance of EKEA, it enhances knowledge transfer efficiency through the transfer mode of mutual exchange and sharing of implicit knowledge. This enhances the impact of EAT on EC. Therefore, we suggest the assumptions below:

H5: EKEA plays a positive regulatory role in the link between EAT and EC, such that strengthening EKEA enhances the positive effect of EAT on EC, while weakening EKEA decreases this impact.

## 3. Methods

### 3.1 Flow chart

3.1 The flow diagram of the methods is displayed below (see Fig 2).

### 3.2 Participants and procedure

This research has received approval from the Hospital Academic Ethics Committee. In testing the hypothesis, this study uses China's medical equipment enterprises as a survey object

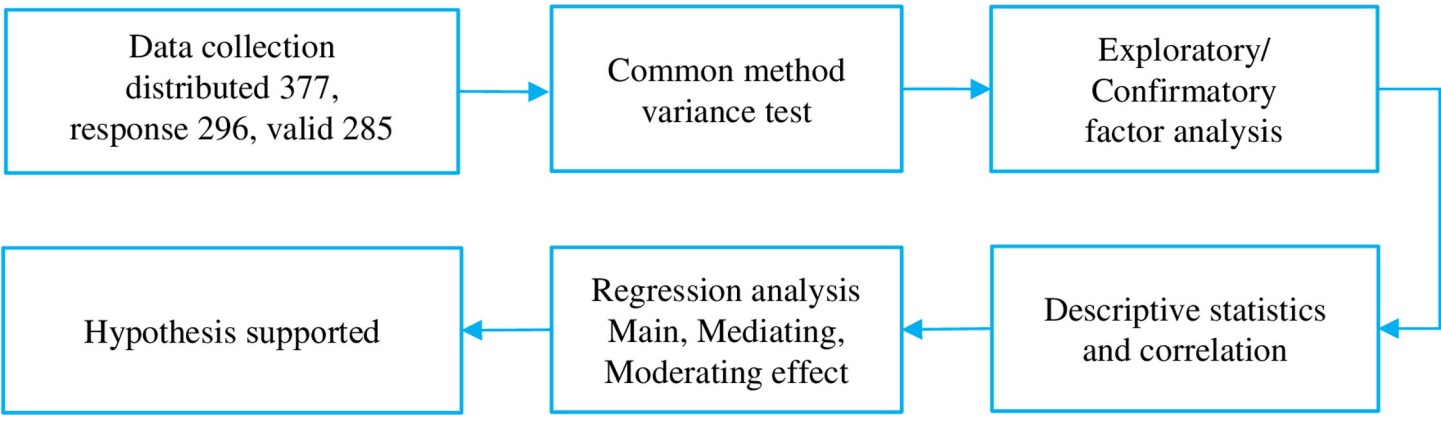

**Fig 2. Flow diagram of the methods.**

through quantitative research methods. In ensuring the clarity of survey questions, focus group discussions with five medical equipment suppliers were undertaken. After several rounds of revisions, a pilot survey was conducted. The pilot survey entailed interviewing 42 senior executives of medical equipment suppliers from two top three hospitals in southwest China. The data analysis results preliminarily verified the study hypothesis.

Furthermore, we obtained a directory of senior executives of medical equipment suppliers from the Alumni platform of several double first-class universities in the southwest region. Through random sampling, we selected more than 400 senior executives from the directory and invited them to participate in the survey via email or phone. After receiving a positive response via email or phone, 377 questionnaires were distributed to senior executives of medical equipment enterprises from March 2023 to April 2023. Finally, we received a total of 296 questionnaires from the senior executives. After deleting invalid questionnaires, the study retained 285 valid samples, representing a participation rate of 75.60%. The characteristics of the respondents are presented in Table 1.

### 3.3 Variables

A seven-point Likert-type was applied to measure the variables, where "1" represents totally disagree and "7" represents totally agree.

**3.3.1 Dependent variable: GIP.** Our approach to measuring GIP was influenced by the gaps identified in previous research [1, 35]. Past studies focused on the innovation process and did not depict what "green" entails. Using the measurement of GIP in previous literature as a reference, this paper redefines and proposes additional measurement indicators of GIP. Based on existing literature [1, 35] and enterprise interview data, this paper measures GIP from four aspects: raw materials, energy conservation, waste disposal, and recyclability. It probed into whether (1) New products or businesses developed by the enterprises focused on the use of environmentally friendly materials; (2) New products or businesses developed by the enterprises attached great importance to energy conservation; (3) The new products or businesses developed by the enterprises reasonably dispose of hazardous substances or waste materials; (4) The new products or businesses developed by the enterprises attach great importance to the recyclability and reusability of the goods sold.

**3.3.2 Independent variable: EAT.** At present, few direct measurements of EAT have been found. Based on similar research [22, 23], this paper measures EAT from four aspects: (1) Enterprise managers actively participate in the environmental protection theoretical training

**Table 1. The features of the respondents.**

| Feature | Category | Quantity | Percentage |
|---|---|---|---|
| Age | < = 30 | 62 | 21.75% |
| | 31–40 | 149 | 52.28% |
| | 41–55 | 58 | 20.35% |
| | > = 56 | 16 | 5.61% |
| Gender | Male | 217 | 76.14% |
| | Female | 68 | 23.86% |
| Education | Undergraduate and below | 103 | 36.14% |
| | Postgraduate and above | 182 | 63.86% |
| Tenure | 0–2 years | 31 | 10.88% |
| | 2–5 years | 39 | 13.68% |
| | 5–10 years | 136 | 47.72% |
| | 10 years and above | 79 | 27.72% |

organized by government agencies or other organizations; (2) Enterprise management personnel actively participate in pro-environmental practice which organized by the government agencies or other organizations; (3) Enterprise managers actively participate in the skills training in low-carbon energy conservation arranged by government institutions or non-governmental organizations; (4) Enterprise managers actively participate in environmental policy-related education and awareness programs, which are arranged by government institutions or non-governmental organizations.

**3.3.3 Mediation variable: Environmental commitment.** With reference to the description of organizational commitment by Meyer and Allen [50] and the measurement indicators of environmental commitment from an individual level presented by He, Cheng [51], this study proposes four components of environmental commitment on an organizational level. These are: (1) Enterprises pay close attention to environmental interests in production and operation; (2) Enterprises' corporate culture and values deeply reflect the performance of environmental protection responsibilities; (3) Enterprises are willing to continuously invest manpower, material resources, and funds to fulfill environmental responsibilities; (4) Enterprises consider environmental protection to be a crucial aspect of its long-term strategy.

**3.3.4 Moderating variable: EKEA.** Drawing on research by Lee and Wong [52] and Kwahk and Park [53], as well as considering the pertinent aspect of environmental protection, this paper proposes four measurement indicators of EKEA, namely: (1) Enterprises actively organize employees to share knowledge and experience beneficial to environmental protection; (2) Enterprises encourage employees to discuss knowledge about environmental protection measures or technologies; (3) Enterprises actively participate in environmental protection related exchange meetings and programs which arranged by government institutions or non-governmental organizations; (4) Enterprises actively participate in the sharing activities of environmental protection related knowledge and technology organized by the industry.

**3.3.5 Control variables.** Factors such as enterprise size, age, ownership, government subsidies, R&D intensity, and net profit can affect the results of enterprise innovation behavior. With reference to research by [54, 55], this paper uses the number of workers to indicate organizational size. The ownership structure of enterprises is measured, as 1 indicates state-owned enterprises and 0 indicates privately owned enterprises [45]. The intensity of R&D is measured by calculating the ratio of R&D investment to total sales.

### 3.4 Reliability and validity

Firstly, we performed a factor analysis to eliminate the influence of common method variance (CMV). The total variance interpretation table shows that the explained variance of the first factor without the pre-rotation eigenvalue greater than 1 is 28.226%, less than 40%. This value, therefore, confirms that there are no serious CMV problems in this study.

Secondly, we used the principal component analysis method to conduct factor analysis on potential variables. The result shows that the dimension division result after factor rotation is the same as the questionnaire setting in this paper. This shows that the potential variable combination corresponding to each variable in this paper is a good fit for the sample distribution characteristics.

Thirdly, we calculated some important observation indicators related to the reliability and validity through SPSS. The results show that all variables had a Cronbach's α reliability coefficient greater than 0.7, demonstrating that the questionnaire responses in this study are highly reliable. Additionally, the KMO value and the variables' factor loading were all above 0.7, indicating that the questionnaire responses are valid and suitable for factor analysis. Furthermore, we found that all composite reliability (CR) value were above 0.7, and the average variance extracted (AVE) were above 0.5. Table 2 depicts that all the observed reliability and validity indicators of this research questionnaire have achieved a satisfactory level.

Furthermore, we carried out a confirmatory factor analysis using AMOS software. The results demonstrate that $\chi^2/df$ value is 1.083, below 3; the RMSEA value is 0.017, below 0.08; the SRMR value is 0.0245, below 0.05. Additionally, key indicators such as CFI, GFI, TLI, IFI, and NFI are 0.997, 0.957, 0.997, 0.997, and 0.967, respectively, exceeding the recommended threshold of 0.9, indicating good structural validity of the questionnaire used in this study.

## 4. Result

### 4.1 Descriptive and correlation analysis

Table 3 describes the sample characteristics and variable correlations. The findings demonstrate that EAT is positively related to GIP, with a coefficient of 0.359, p<0.01, and EC, with a

**Table 2. Reliability & validity test.**

| Variables | Cronbach's α | AVE | CR | KMO | Factor loading |
|---|---|---|---|---|---|
| Environmental awareness training | 0.909 | 0.785 | 0.9359 | 0.851 | 0.878 |
| | | | | | 0.888 |
| | | | | | 0.893 |
| | | | | | 0.885 |
| Environmental Commitment | 0.927 | 0.8205 | 0.9481 | 0.854 | 0.892 |
| | | | | | 0.912 |
| | | | | | 0.906 |
| | | | | | 0.913 |
| Environmental knowledge exchange activities | 0.909 | 0.7870 | 0.9366 | 0.846 | 0.906 |
| | | | | | 0.874 |
| | | | | | 0.896 |
| | | | | | 0.872 |
| Green innovation performance | 0.900 | 0.7696 | 0.9304 | 0.850 | 0.866 |
| | | | | | 0.873 |
| | | | | | 0.880 |
| | | | | | 0.890 |

**Table 3. Mean, SD and correlation.**

| Variables | Size | Age | State-Ownership | Financial Subsidy | R&D Intensity | Net profits | EAT | EC | EKEA | GIP |
|---|---|---|---|---|---|---|---|---|---|---|
| Size | 1 | | | | | | | | | |
| Age | .253** | 1 | | | | | | | | |
| State-Ownership | .056 | .151* | 1 | | | | | | | |
| Financial Subsidy | .156** | .209** | .122* | 1 | | | | | | |
| R&D Intensity | .127* | -.062 | .059 | .216** | 1 | | | | | |
| Net profits | .260** | -.021 | .202** | .090 | .182** | 1 | | | | |
| EAT | .263** | .025 | .172** | .107 | .325** | .310** | 1 | | | |
| EC | .249** | .006 | .222** | .174** | .177** | .348** | .457** | 1 | | |
| EKEA | .009 | .015 | -.045 | .023 | .071 | .106 | -.041 | .030 | 1 | |
| GIP | .163** | -.027 | .182** | .242** | .396** | .22** | .359** | .328** | -.017 | 1 |
| Mean | 2.400 | 2.614 | 0.372 | 2.867 | 2.768 | 2.811 | 3.904 | 3.964 | 4.159 | 3.941 |
| SD | 1.377 | 1.345 | 0.484 | 1.442 | 1.415 | 1.480 | 1.737 | 1.750 | 1.743 | 1.696 |

Note: N = 285;

* $p < 0.05$,

** $p < 0.01$.

coefficient of 0.457, $p < 0.01$. Furthermore, EC is positively related to GIP, with a coefficient of 0.328, $p < 0.01$. This is consistent with the fundamental hypothesis of this study. At the same time, all regression models in this paper have VIFs less than 10, indicating the absence of severe collinearity.

## 4.2 Hypotheses testing

This paper utilized SPSS and Process software to examine the correlation between variables and verify the proposed research hypothesis.

**4.2.1 Direct effect and mediation effect.** Table 4 presents the outcomes of the regression analysis of both the direct and mediation effects.

**Table 4. Regression analysis—main effect and mediation effect.**

| Variables | GIP: M1—M4 | | | | EC: M5—M6 | |
|---|---|---|---|---|---|---|
| | **M1** | **M2** | **M3** | **M4** | **M5** | **M6** |
| 1.Size | 0.084(0.070) | 0.047(0.07) | 0.051(0.07) | 0.032(0.07) | 0.168(0.074) ** | 0.106(0.071) |
| 2.Age | -0.077(0.072) | -0.074(0.07) | -0.063(0.07) | -0.064(0.07) | -0.072(0.075) | -0.066(0.071) |
| 3. State-Ownership | 0.131(0.192) * | 0.108(0.189) * | 0.100(0.191) | 0.091(0.189) | 0.156(0.202) ** | 0.117(0.191) * |
| 4. Financial Subsidy | 0.151(0.066) ** | 0.152(0.064) ** | 0.13(0.065) * | 0.136(0.064) * | 0.105(0.069) | 0.107(0.065) * |
| 5. R&D Intensity | 0.322(0.066) *** | 0.268(0.067) *** | 0.307(0.065) *** | 0.27(0.067) *** | 0.074(0.07) | -0.017(0.068) |
| 6. Net profits | 0.103(0.065) | 0.062(0.065) | 0.053(0.066) | 0.036(0.066) | 0.248(0.068) *** | 0.181(0.066) ** |
| 7.EAT | | 0.208(0.057) *** | | 0.158(0.06) * | | 0.348(0.057) *** |
| 8.EC | | | 0.198(0.056) *** | 0.145(0.059) * | | |
| R2 | 0.210 | 0.242 | 0.240 | 0.255 | 0.177 | 0.272 |
| F | 13.585*** | 13.964*** | 13.788*** | 13.131*** | 11.146*** | 16.127*** |

Note:

***$p < 0.001$,

**$p < 0.01$,

*$p < 0.05$.;N = 285.

M1 depicts the effect of the selected control variables on GIP. The findings indicate that the ownership, financial subsidies and proportion of R&D investment to sales revenue are positively associated with GIP. This is in line with existing research conclusions.

M2 depicts the regression analysis results on the impact of the explanatory variable EAT on GIP while controlling for the influence of other variables. The findings indicate that the variable EAT positively influences GIP($\beta = 0.208$, $p<0.001$), thereby supporting H1.

M4 shows the regression analysis findings on the impact of the explanatory variable EC on GIP while controlling for the influence of other variables. The results indicate a significant positive effect of EC on GIP ($\beta = 0.145$, $p<0.05$), which confirms H3.

M6 examines the regression analysis on the impact of the explanatory variable, EAT on EC while controlling for the influence of other variables. The findings indicate a significant positive impact of EAT on EC($\beta = 0.348$, $p<0.001$), which supports H2.

Based on the findings of M1-M6, we can preliminarily speculate the mediating role of EC in the relationship between EAT and GIP according to the basic meaning of mediating variables. To assess the strength and validity of the mediating effect, we further conducted a Bootstrap analysis using the PROCESS software. The findings of the Bootstrap analysis are depicted in Table 5. The results indicate a significant mediation effect of EC on the relationship between EAT and GIP. The 95% confidence interval does not include 0, suggesting the results are robust and reliable. The direct effect value was 0.1538, accounting for 75.76% of the total effect, while the value of the indirect effect was 0.0492, representing 24.24% of the total effect. These findings are in support of H4.

**4.2.2 Moderating effect test.** Table 6 presents the findings of the regulatory effect model's regression analysis. To mitigate the impact of collinearity, we first decentralized interaction terms between the independent and regulatory variables. M9 demonstrates a significant positive impact of the interaction effect between EAT and EKEA on EC ($\beta = 0.180$, $p<0.001$). These findings are in support of H5.

In order to better demonstrate the impact of moderating variables, we subsequently used the PROCESS software to draw a moderating effect diagram and got Fig 3. Fig 3 shows that when the level of EAT is held constant, an increase in EKEA will significantly enhance the level of EC. These findings demonstrate a significant moderating effect of EKEA and support H5 again.

**4.2.3 Moderated mediation effect.** In exploring the moderated mediation effect, we use the PROCESS software to run MODEL 7 (here, select MODEL 7 in the software) to calculate the data. The findings in Table 7 depict that the conditional indirect effect of EKEA on EAT-EC-GIP was significant. Moreover, the 95% confidence intervals for the conditional indirect effect did not include 0, indicating a significant moderated mediation effect. The coefficient of the moderated mediation effect is 0.0146, and the confidence interval does not include 0, indicating that the Moderated mediation effect of EKEA between EAT-EC-GIP is significant.

**Table 5. Bootstrap mediating effect test.**

| IV | DV: GIP | | | | | |
|---|---|---|---|---|---|---|
| | Types | Effect | Boot SE | Bootstrap 95% CI | | Rate |
| | | | | LLCI | ULCI | |
| EAT | Total | 0.2030 | 0.0568 | 0.0912 | 0.3147 | 100% |
| | Direct | 0.1538 | 0.0600 | 0.0358 | 0.2719 | 75.76% |
| | Indirect | 0.0492 | 0.0216 | 0.0116 | 0.0982 | 24.24% |

**Table 6. Regression analysis—regulation effect.**

| Variable | EC | | |
| --- | --- | --- | --- |
| | **M7** | **M8** | **M9** |
| 1. Size | 0.168(0.074) ** | 0.107(0.071) | 0.095(0.070) |
| 2. Age | -0.072(0.075) | -0.068(0.071) | -0.057(0.070) |
| 3. State-Ownership | 0.156(0.202) ** | 0.119(0.192) * | 0.106(0.189) * |
| 4. Financial Subsidy | 0.105(0.069) | 0.107(0.065) * | 0.116(0.064) * |
| 5. R&D Intensity | 0.074(0.070) | -0.019(0.069) | -0.036(0.067) |
| 6. Net profits | 0.248(0.068) *** | 0.177(0.066) ** | 0.174(0.065) ** |
| EAT | | 0.351(0.058) *** | 0.34(0.057) *** |
| IKEA | | 0.03(0.052) | 0.017(0.051) |
| EAT*EKEA | | | 0.180(0.029) *** |
| R2 | 0.177 | 0.270 | 0.299 |
| F | 11.146*** | 14.121*** | 14.483*** |

Note:

***p<0.001,

**p<0.01,

*p<0.05.;N = 285.

# 5. Conclusion, implications, and limitation

## 5.1 Conclusion

Knowledge of environmental awareness is crucial in today's business practices, growth, and sustained competitive edge. Environmental awareness training offers corrective measures to mitigate ecological menace towards innovation performance. Although research has uncovered the significance of pro-environmental education on environmental protection activities, the impact of environmental awareness training on green innovation performance remains underexplored, especially in emerging economies. This study fills the literature gap by using

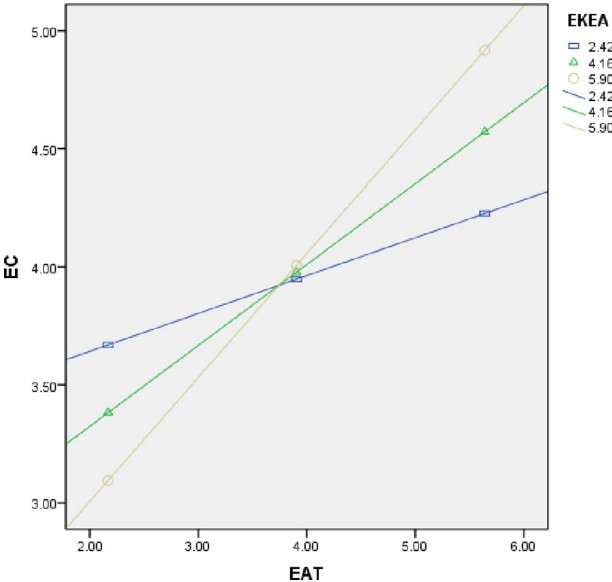

**Fig 3. The moderating effect of EKEA.**

**Table 7. Moderated mediation effect.**

| IV | Moderating variable | EAT→EC→GIP | | | | | | | |
|---|---|---|---|---|---|---|---|---|---|
| | | Conditional indirect effect | | | | Moderated mediation effect | | | |
| | | Effect | Standard error | LLCI | ULCI | Effect | Standard error | LLCI | ULCI |
| EAT | Low EKEA | 0.0225 | 0.0149 | 0.0008 | 0.0621 | 0.0146 | 0.0075 | 0.0033 | 0.0336 |
| | High EKEA | 0.0735 | 0.0318 | 0.0157 | 0.1402 | | | | |

survey data from 285 medical equipment manufacturing companies in China to investigate the mechanism of EAT's influence on GIP, the mediating role of EC and the positive regulatory impact of EKEA, and the logical relationship between these variables from the perspective of KBV. The results show that: (1) There is a positive relationship between EAT and GIP; (2) EAT positively affects EC; (3) EC positively affects GIP; (4) EC plays a mediation role between EAT and GIP; (5) EKEA positively regulates the relationship between EAT and EC.

## 5.2 Theoretical implications

Firstly, this paper innovatively developed an indicator for measuring the variables of EAT and GIP, thereby contributing to the methodological advancement of research in this area. The measurement of GIP by previous studies [1, 35] focused on the "innovation process" and did not depict what "green" entails. Using the measurement of GIP in previous literature as a reference, this paper redefines and proposes additional measurement indicators of GIP. This study thus measures GIP from four facets: raw materials, energy conservation, waste disposal, and recyclability. Presently, little literature on the direct measurement of EAT has been found. Based on similar research [22, 23], this paper designs EAT measurement indicators from four perspectives namely: theoretical learning, practical works, skills training, and environmental protection policies. The conceptual framework and measurement results used comprehensively consider existing theories and insights from organizational research. The high reliability of these measures enhances their value as a reference for similar studies in the future.

Second, we have contributed to research related to EAT and GIP. Our empirical analysis reveals that EAT has a positive influence on GIP. This conclusion is somewhat in line with prior academic works such as Xie, Chen [3, 4], which highlight the pivotal role that pro-environmental education plays in influencing environmental protection activities. Our study adds nuance to the extant works in several key ways. Foremost among these is the targeted focus on the impact of EAT on GIP, which has hitherto been underexplored, particularly within the medical equipment manufacturing sector in emerging economies. Our results extend KBV by demonstrating that it plays a key function in the impact of EAT on GIP.

Third, this research depicts the internal influence mechanism of EAT on GIP from the perspective of KBV. Existing research has conducted extensive research on GIP based on methods [56, 57], knowledge management [58], quality management [59], green core competence [29], and other aspects. The current research neglects the influence of the managerial subjective perception. The environmental education and training for enterprise managers will inevitably affect their objective conscious behavior and values. Research on internal influence mechanisms is needed to attain a holistic understanding of EAT. This paper reveals the internal process mechanism of "Environmental awareness training—Environmental commitment—Green innovation performance" based on the KBV perspective, filling this research gap, which is somewhat in line with prior academic works such as Mo, Boadu [1] and Xie, Chen [4] that highlight an interfering consequence of managerial environmental concerns in the correlation between pro-environmental education and eco-friendly agricultural production and CSR

activities and GIP, respectively. Our study delves deeper by identifying that EC mediates the correlation between EAT and GIP. This indicates that EAT and EC are key stepping stones for the GIP to take place effectively. The study is an add-on to KBV and related studies on GIP.

Finally, utilizing the knowledge-based view as a theoretical framework, this research analyzed the moderating effect of EKEA and identified a boundary condition that enhances the impact of EAT on EC. EAT by government organizations promotes the acceptance of relevant environmental protection concepts, thus effectively promoting EC. However, in this process, most of the knowledge is explicit knowledge, ignoring the implicit process of explicit knowledge [60], which ultimately leads to problems such as low efficiency and low availability of EAT knowledge in the transmission process and acceptance. Based on this, we introduced the variable EKEA, which can help enterprise executives solve challenges that arise during the knowledge transfer process [61]. From the knowledge-based perspective, this paper examines the reinforcement of EKEA on EAT and EC and the moderated mediation effect of EKEA on "Environmental awareness training—Environmental commitment—Green innovation performance." These findings contribute to extending the scope of the KBV in environmental education.

## 5.3 Management implications

There are three main management implications of this study:

Firstly, the research findings indicate that EAT is significantly associated with GIP. We suggest civil society organizations and government agencies organize environmental protection awareness, sensitization, and training programs for enterprise managers to grip EAT concepts in their green innovation policies.

Secondly, the results of this study show that EKEA moderates the relationship between EAT and EC and plays a significant role in regulating the overall process of "Environmental awareness training—Environmental commitment—Green innovation performance." This shows that environmental knowledge exchange activities are conducive to managers' assimilation, accelerate the knowledge-gathering process, and thus improve the efficiency of knowledge transmission and reception. We suggest that enterprises consider EKEA as a process in the publicity activities of environmental protection knowledge. The exchange of training experiences among participants after the training sessions can aid in improving managers' green innovation awareness, thus promoting EC.

Finally, authorities or government departments should formulate relevant incentive measures, implement the green development policy, and integrate environmental protection awareness and sensitization into their strategic framework. And apply it to develop the capacity of enterprise managers, which, in the long run, improves the GIP of businesses.

## 5.4 Limitation

Though this research has made significant contributions to related knowledge, several limitations should be noted.

First of all, concerning the sample and scope of the study, data was only collected on management at the enterprise level. However, factors such as enterprise grassroots implementation, government policies, economic environment, and social recognition are likely to affect GIP. Future studies can, therefore, probe into these factors in assessing the relationship between environmental awareness training and green innovation performance. The multiple perspectives from these findings will ensure more accurate and objective conclusions.

The second limitation of this study is the variable design. In the design process of indicators, the study considered both subjective and objective variables provided in the literature.

The subjective nature of some variables introduced, such as EKEA and EC, may result in biases and erroneous conclusions that require further empirical analysis. Subsequent research can, therefore, reassess these variables based on relevance to improve the research framework and obtain more accurate research conclusions.

Finally, this study was conducted in a single geographic context (China) in a single sector (medical equipment manufacturing companies in China). Therefore, researchers must be cautious when generalizing these results and conclusions to other settings. Replicating this study in a different geographic context or sector would be helpful to generalize our insights and conclusions.

## Acknowledgments

The authors would like to thank the editors and anonymous reviewers for their efforts in improving our paper.

## Author Contributions

**Conceptualization:** Qiong Wu, Zhen Chen.

**Data curation:** Qiong Wu, Senlin Xie, Zhen Chen.

**Funding acquisition:** Zhen Chen.

**Investigation:** Qiong Wu, Senlin Xie, Zhen Chen.

**Methodology:** Qiong Wu, Senlin Xie, Zhen Chen.

**Supervision:** Zhen Chen.

**Validation:** Qiong Wu, Senlin Xie.

**Writing – original draft:** Qiong Wu, Senlin Xie, Lucille Aba Abruquah, Zhen Chen.

**Writing – review & editing:** Qiong Wu, Senlin Xie, Shihan Wang, Anning Zhou, Lucille Aba Abruquah, Zhen Chen.

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
