## [Decision Letter · Decision Letter 0]

26 Jul 2023

PONE-D-23-15337Effects of environmental awareness training and environmental commitment on firm’s green innovation performance: Empirical insights from medical device suppliersPLOS ONE

Dear Dr. Chen,

Thank you for submitting your manuscript to PLOS ONE. After careful consideration, we feel that your manuscript has merit but does not fully meet PLOS ONE’s publication criteria as it currently stands. Therefore, we invite you to submit a revised version of the manuscript that addresses the points raised during the review process. Please submit your revised manuscript by Sep 09 2023 11:59PM. If you will need more time than this to complete your revisions, please reply to this message or contact the journal office at plosone@plos.org. Please include the following items when submitting your revised manuscript:A rebuttal letter that responds to each point raised by the academic editor and reviewer(s). You should upload this letter as a separate file labeled 'Response to Reviewers'.A marked-up copy of your manuscript that highlights changes made to the original version. You should upload this as a separate file labeled 'Revised Manuscript with Track Changes'.An unmarked version of your revised paper without tracked changes. You should upload this as a separate file labeled 'Manuscript'.

We look forward to receiving your revised manuscript.

Kind regards,

Jibril Adewale Bamgbade

Academic Editor

PLOS ONE

Journal Requirements:

"This work was supported by the National Natural Science Foundation of Chongqing [Grant Number CSTB2022NSCQ-MSX0502]; and the City-School Cooperation Project of Dazhou Science and Technology Bureau [Grant Number DZKJJ2020S05]. The entire process and conclusions of this study were not influenced by the fund sponsor."

Reviewers' comments:

Reviewer's Responses to Questions

**Comments to the Author**

1. Is the manuscript technically sound, and do the data support the conclusions?

Reviewer #1: Partly

Reviewer #2: Yes

2. Has the statistical analysis been performed appropriately and rigorously? 

Reviewer #1: Yes

Reviewer #2: Yes

3. Have the authors made all data underlying the findings in their manuscript fully available?

Reviewer #1: Yes

Reviewer #2: Yes

4. Is the manuscript presented in an intelligible fashion and written in standard English?

Reviewer #1: Yes

Reviewer #2: Yes

5. Review Comments to the Author

Reviewer #1: In summary, although this paper provides valuable insights in examining the effect of environmental awareness training and environmental commitment on firm’s green innovation performance,more work is needed to expand on the literature review, improve the paper structure and writing, and deepen the theoretical contributions.

Reviewer #2: The article is interesting and describes an important research problem. However, I recommend making a few changes and additions.

1. The abstract should not repeat what has been said in the research background. The first two paragraphs in the research background part are the same as in the abstract. So, you need to rewrite the research background part to not repeat the abstract.

2. Discussion should be included in the main body of the paper. More in-depth discussion should be included to support the interpretations and conclusions. You should start with an overall comment about the findings and then critically evaluate the main issues raised by the study. You should also present a coherent argument for their position.

3. You should clearly present all the measurement items of variables (e.g. in a table of Appendix).

6. PLOS authors have the option to publish the peer review history of their article (what does this mean?). If published, this will include your full peer review and any attached files.

Reviewer #1: No

Reviewer #2: No

---

## [Author Response · Author response to Decision Letter 0]

14 Sep 2023

Dear Editor,

Thanks for your encouragement and the opportunity to revise and resubmit our manuscript to the PLOS. We have successfully addressed all the comments and presented an improved paper version. Thank you again.

Review Comments & Respond.

Reviewer #1: In summary, although this paper provides valuable insights in examining the effect of environmental awareness training and environmental commitment on firm’s green innovation performance, more work is needed to expand on the literature review, improve the paper structure and writing, and deepen the theoretical contributions.

Respond: Thanks for your professional review and modification suggestions. We have successfully addressed your comments and presented an improved paper version. First, we have improved the paper structure and writing (see the edited paper.) Second, we have revised the literature review and deepened the theoretical contributions. It reads:

2. Literature Review

2.1 Knowledge-Based View

We draw from the knowledge-based view (KBV) to explore the relationship between environmental awareness activities and GIP. The KBV of the enterprise builds and extends the resource-based view (RBV) that emphasizes the strategic significance of knowledge as a unique resource (13). The theory concentrates on how enterprises create, acquire, protect, transfer, and use knowledge (14) for higher performance. Thus, it is regarded as the convergence of the enterprise knowledge research flow. Proponents of KBV perceive knowledge as the most significant strategic organizational resource in terms of value creation. Although, lack of consensus over the nature of knowledge has prevented the KBV from developing into an integrated, stand-alone theory, but has not constrained its ability to provide penetrating insights into organizational strategy and management. Boadu, Xie (15) assert that the fundamental purpose of an organization is to create and apply knowledge that can offer the foundation for sustainable differentiation (i.e., difficult to imitate) to attain a competitive edge. The theory asserts that knowledge base and ability are the main determinants for enterprises to obtain results in the knowledge economy (16). Though multifaceted, scholars have established that the amalgamation and the configuration of intangible resources, such as knowledge resources, are crucial to innovation (4). Admittedly, KBV postulates that focusing on the efficiency of knowledge exchange and searching for the best method can boost the management and operation of business activities, leading to organizational innovation and competitive edge (14, 17). Hence, the KBV perspective is suitable for scrutinizing environmental awareness knowledge exchange and GIP. It explicitly addresses how EAT affects the knowledge base of managers, which is reflected in the management's future business decisions (18), resulting in an enhanced EC and ultimately improving the GIP of enterprises. 

5.2 Theoretical implications

Firstly, this paper innovatively developed an indicator for measuring the variables of EAT and GIP, thereby contributing to the methodological advancement of research in this area. The measurement of GIP by previous studies (1, 35) focused on the "innovation process" and did not depict what "green" entails. Using the measurement of GIP in previous literature as a reference, this paper redefines and proposes additional measurement indicators of GIP. This study thus measures GIP from four facets: raw materials, energy conservation, waste disposal, and recyclability. Presently, little literature on the direct measurement of EAT has been found. Based on similar research (22, 23), this paper designs EAT measurement indicators from four perspectives namely: theoretical learning, practical works, skills training, and environmental protection policies. The conceptual framework and measurement results used comprehensively consider existing theories and insights from organizational research. The high reliability of these measures enhances their value as a reference for similar studies in the future. 

Second, we have contributed to research related to EAT and GIP. Our empirical analysis reveals that EAT has a positive influence on GIP. This conclusion is somewhat in line with prior academic works such as Xie, Chen (4) and (3), which highlight the pivotal role that pro-environmental education plays in influencing environmental protection activities. Our study adds nuance to the extant works in several key ways. Foremost among these is the targeted focus on the impact of EAT on GIP, which has hitherto been underexplored, particularly within the medical equipment manufacturing sector in emerging economies. Our results extend KBV by demonstrating that it plays a key function in the impact of EAT on GIP. 

Third, this research depicts the internal influence mechanism of EAT on GIP from the perspective of KBV. Existing research has conducted extensive research on GIP based on methods (56, 57), knowledge management (58), quality management (59), green core competence (29), and other aspects. The current research neglects the influence of the managerial subjective perception. The environmental education and training for enterprise managers will inevitably affect their objective conscious behavior and values. Research on internal influence mechanisms is needed to attain a holistic understanding of EAT. This paper reveals the internal process mechanism of "Environmental awareness training—Environmental commitment—Green innovation performance" based on the KBV perspective, filling this research gap, which is somewhat in line with prior academic works such as Xie, Chen (4) and Mo, Boadu (1) that highlight an interfering consequence of managerial environmental concerns in the correlation between pro-environmental education and eco-friendly agricultural production and CSR activities and GIP, respectively. Our study delves deeper by identifying that EC mediates the correlation between EAT and GIP. This indicates that EAT and EC are key stepping stones for the GIP to take place effectively. The study is an add-on to KBV and related studies on GIP.

Finally, utilizing the knowledge-based view as a theoretical framework, this research analyzed the moderating effect of EKEA and identified a boundary condition that enhances the impact of EAT on EC. EAT by government organizations promotes the acceptance of relevant environmental protection concepts, thus effectively promoting EC. However, in this process, most of the knowledge is explicit knowledge, ignoring the implicit process of explicit knowledge (60), which ultimately leads to problems such as low efficiency and low availability of EAT knowledge in the transmission process and acceptance. Based on this, we introduced the variable EKEA, which can help enterprise executives solve challenges that arise during the knowledge transfer process (61). From the knowledge-based perspective, this paper examines the reinforcement of EKEA on EAT and EC and the mediation effect of EKEA on “Environmental awareness training—Environmental commitment—Green innovation performance.” These findings contribute to extending the scope of the KBV in environmental education. 

Attachment review comments & respond:

Comments:

The basic idea of the paper is probably nice, The aim of this paper explores the cause and effect relationship of environmental awareness training (EAT), environmental knowledge exchange activities (EKEA), and environmental commitment (EC) on green innovation performance (GIP).Research investigating the effectiveness of environmental education programs and initiatives aimed at promoting the exchange of environmental knowledge. To bridge the research gap, this study utilized the survey data from Chinese medical device manufacturers to verify the research assumptions. Focus group discussions with five medical equipment suppliers were undertaken, and survey data from 285 medical equipment manufacturing companies in China analyzed with a confirmatory factor analysis using AMOS software techniques. Lastly, paper utilized SPSS and Process software to examine the correlation between variables and verify the proposed research hypothesis.

The paper shows excellence in novelty and originality, and the findings have important implications for environmental awareness training that plays a crucial role in promoting green innovation performance,and are clearly presented with assurance of correctness. Also, the study has the potential to influence further research in the field of environmental awareness training and innovation performance. However, there are still some shortcomings. Specific comments are listed below.

1. Introduction

The introduction has some strengths, such as providing a clear overview of the topic and highlighting the significance of green innovation performance (GIP).The study is unable to come up with a specific problem from the introduction part. However, there are some weaknesses in the structure and writing of the introduction.

- Firstly, the introduction could be more concise and focused, it could be trimmed down to make the introduction more efficient. Secondly, the introduction could benefit from clearer transitions between ideas. Some of the paragraphs feel disconnected from each other, which makes it harder for the reader to follow the argument. 

- Finally, there could be more attention to the flow of the introduction. It might be more effective to start by introducing the problem of GIP firm adoption and the resistance it faces before going into the background information. This would make the introduction more engaging and help to create a sense of urgency around the research question, the introduction could be more effectively set up the rest of the paper.

Respond: Thanks for your comment and good suggestion. We appreciate your suggestions and have modified the paper along the line. The current version has undergone significant improvements. Thank you once again for your advice. It reads:

1. Research background

The significance of green innovation cannot be overemphasized in developed and emerging economies. It plays a critical function in the twenty-first century for enterprises’ success, development, and survival. Mo, Boadu (1) assert that green innovation manages the environment, energy usage, pollutant production, and waste disposal and recycling towards cleaner production and sustainability. Thus, it serves as a superior player, mitigates the production of pollutants, and increases cost efficiency and competitive advantage. For the past few years, stakeholders have gradually become aware of green innovation’s importance to society and how enterprises should operate in an environmentally friendly manner to integrate green practices into strategic and process management. However, compared with developed markets environment, emerging markets face challenges between development and the environment. For instance, with the escalation of global industrialization, Chinese enterprises' production and operating activities have caused pollution, threatening the life of the earth's ecosystem (2). Due to the alarming nature of the menace, for enterprises to survive, scholars, government agencies, and civil society organizations have recognized environmental protection sensitization and awareness training programs as a vital weapon for curbing the rising environmental issues to promote sustainability. Environmental awareness training refers to the sum of the training and learning actions on ecological issues, hands-on activities for sustainability, and relevant environmental policies and regulations taken by enterprises to meet environmental challenges. Environmental awareness training activities offer corrective measures to mitigate ecological menace towards innovation performance. Previous studies have established the positive impact of pro-environmental education on environmental protection (3, 4). However, there is a dearth of studies on the environmental awareness training effectiveness on the green innovation performance (GIP) of enterprises, especially in emerging economies. This research aims to bridge the gap by exploring the cause-and-effect relationship between environmental awareness training (EAT), environmental knowledge exchange activities (EKEA), and environmental commitment (EC) in GIP and related boundary conditions using the Chinese medical equipment manufacturing industry. Chinese market environment offers a fertile ground for the study of such nature to understand the linkage amongst the variables. 

First, research shows that education and training experiences are crucial in business management decisions (5, 6). For instance, extant works have established the positive impact of pro-environmental education on environmental protection (3, 4). However, there is still a lack of relevant evidence on the specific extent to which environmental training impacts GIP. Hence, the current study probes the linkage between EAT and GIP. 

Second, the scholarship contends that EC mediates between EAT and GIP. From the extant literature, research on EC primarily centers on the basic level of individuals and the environment. For example, Hojnik, Ruzzier (7) explore the determinants of individuals' inclination to pay for green energy. Cicatiello, Ercolano (8) explore the relationship between personal views on the environment and the environment of a specific region. Few studies have, however, linked EC with GIP to examine the impact of EC on EAT and GIP. This research thus investigates the involvement and efficacy of EC in the green innovation process of medical equipment enterprises to bridge the research gap. 

Third, leveraging the knowledge-based view, this research probes into the expansion effect of EKEA on the relationship between EAT and EC. EKEA can aid enterprises in overcoming knowledge limitations that arise within a single entity (9), promote collaboration (10) and multilevel knowledge sharing (11), and effectively improve the green innovation performance of enterprises (12). Hence, this scholarship examines the moderating effect of EKEA in the model and expands the application of a knowledge-based view in environmental protection-related fields. 

In summary, to solve the above problems, this scholarship uses survey data from 285 medical equipment manufacturing companies in China to investigate the mechanism of EAT's influence on GIP, the mediating role of EC and the positive regulatory impact of EKEA, and the logical relationship between these variables from the perspective of knowledge-based theory. The findings are expected to expand the applications of EKEA and EC in the green innovation field, making it an essential contribution to the theoretical and practical aspects of the subject. Thus, it provides the academic community, enterprises, and governments with a further understanding of the role of EAT, EC, and EKEA in green innovation planning. It also provides a decision-making reference for the green development of medical equipment manufacturing enterprises. 

2. Literature Review and Hypotheses Development

- The title of the second part should be "Literature Review" instead of "Theory and hypotheses development".

- The paper proposes using the Knowledge Based View (KBV) as an underpinning theory for the study. While KBV is a comprehensive framework, the introduction does not specify the limitations of the model, and whether it has been sufficiently validated in the context of GIP firms.

- The paper proposes investigating environmental awareness training, environmental knowledge exchange activities, and environmental commitment along with the functional elements of KBV. While these are important aspects to consider, the introduction does not provide a clear rationale for why these dimensions are critical, and how they integrate with KBV to create a more robust model for this study. However, the introduction does not explain the rationale for why this integration is necessary or how it contributes to the research objectives.

- Does the paper demonstrate an adequate understanding of the relevant literature in the field and cite an appropriate range of literature sources?Is any significant work ignored? The authors does not provide an overview of the existing literature, which could be more systematic. Even thou am somehow a supporter of tabular literature reviews that illustrate the respective research focuses. This could also be done here if authors feel like. All in all, there has been much more research on GIP than what is presented here. Authors might think about integrating more research perspectives:It appears suitable overall, but the literature is then very focused on it.

Respond: Thanks for your good suggestion. We have changed the sub-titled from “theory and hypothesis development” to “literature review.” Besides, we have addressed the issues concerning the theory (KBV) adopted for the study, especially its rationale and limitations (e.g., although lack of consensus over the nature of knowledge has prevented the KBV from developing into an integrated, stand-alone theory, but has not constrained its ability to provide penetrating insights into organizational strategy and management). Finally, the authors have incorporated more information on green innovation performance. Thank you once again for your suggestion. It read:

2. Literature Review

2.1 Knowledge-Based View

We draw from the knowledge-based view (KBV) to explore the relationship between environmental awareness activities and GIP. The KBV of the enterprise builds and extends the resource-based view (RBV) that emphasizes the strategic significance of knowledge as a unique resource (13). The theory concentrates on how enterprises create, acquire, protect, transfer, and use knowledge (14) for higher performance. Thus, it is regarded as the convergence of the enterprise knowledge research flow. Proponents of KBV perceive knowledge as the most significant strategic organizational resource in terms of value creation. Although, lack of consensus over the nature of knowledge has prevented the KBV from developing into an integrated, stand-alone theory, but has not constrained its ability to provide penetrating insights into organizational strategy and management. Boadu, Xie (15) assert that the fundamental purpose of an organization is to create and apply knowledge that can offer the foundation for sustainable differentiation (i.e., difficult to imitate) to attain a competitive edge. The theory asserts that knowledge base and ability are the main determinants for enterprises to obtain results in the knowledge economy (16). Though multifaceted, scholars have established that the amalgamation and the configuration of intangible resources, such as knowledge resources, are crucial to innovation (4). Admittedly, KBV postulates that focusing on the efficiency of knowledge exchange and searching for the best method can boost the management and operation of business activities, leading to organizational innovation and competitive edge (14, 17). Hence, the KBV perspective is suitable for scrutinizing environmental awareness knowledge exchange and GIP. It explicitly addresses how EAT affects the knowledge base of managers, which is reflected in the management's future business decisions (18), resulting in an enhanced EC and ultimately improving the GIP of enterprises. 

2.2 Environmental Awareness Training and Green Innovation Performance

Green innovation performance measures the extent to which organizations develop innovations that condense or abate environmental damages, impact, and worsening while optimizing the use of natural resources (1). Thus, it denotes the innovative practices of businesses in products, processes, management, and reduction of pollution emissions. As a strategy, it offers great opportunities to meet buyers’ requirements while preserving the ecosystem (1). Admittedly, the concept offers organizations a roadmap for achieving sustainable competitive advantages in an ecologically effective way (1, 19).

3. Methodology

-Is the paper's argument built on an appropriate base of theory, concepts, or other ideas? Has the research or equivalent intellectual work on which the paper is based been well designed? Are the methods employed appropriate?: The hypothesis development is rather limited. 

- There is a lack of a truly informed reflection on the theories. All in all, there is no reference to young consumers in the entire theoretical foundation. The fact that the empirical study itself was conducted with "young tourists" is not sufficient.

- What sample techniques does the authors use for this study, the entire sampling process, as well as the data cleaning process, are somewhat opaque.

Respond: Thanks for your comment and good suggestion. We have enhanced our hypothesis development. We used KBV as a theoretical foundation to develop our hypotheses. The theory serves as a guide to explain the various variables of the study. Besides, the study adopted a random sampling technique to select the elements for the investigation. We have improved the entire sampling process. It reads:

2.2 Environmental Awareness Training and Green Innovation Performance

Green innovation performance measures the extent to which organizations develop innovations that condense or abate environmental damages, impact, and worsening while optimizing the use of natural resources (1). Thus, it denotes the innovative practices of businesses in products, processes, management, and reduction of pollution emissions. As a strategy, it offers great opportunities to meet buyers’ requirements while preserving the ecosystem (1). Admittedly, the concept offers organizations a roadmap for achieving sustainable competitive advantages in an ecologically effective way (1, 19). From the extant studies, green innovation has been categorized into two aspects, namely, product and process innovation (20). However, the ultimate goal is to condense environmental risks in product and service functions for customers, resource utilization and cost efficiency, and organizational flexibility (21). 

From the extant works, there is no clear definition of EAT. This paper draws from similar research on environment protection and training (22, 23) to define EAT as the sum of the training and learning actions on environmental issues, hands-on activities for sustainability, and relevant environmental policies and regulations taken by enterprises to meet environmental challenges. Drawing on the KBV, EAT can influence the manager's knowledge and environmental awareness by facilitating the exchange and learning of relevant knowledge, leading to improved green innovation performance (24, 25). It is predominantly manifest in two main ways: First, it allows managers to acquire knowledge associated with preserving the environment (e.g., environmental concepts, culture, and values). Thus, EAT is more inclined towards green innovation-related strategies and actions in management decisions, which can promote the green innovation performance of enterprises (26, 27). Second, there is evidence to suggest that organizational identity has a positive impact on GIP (28). Previous studies have established the positive impact of pro-environmental education on environmental protection (4).

In this context, we contend that training on environmental protection in the Chinese medical equipment manufacturing sector can promote managers' willingness to participate in environmental protection activities, which, in turn, enhances the green corporate image and identity of enterprises and ultimately promotes GIP (29). For this, we suggest the following hypothesis:

H1: Environmental awareness training has a positive impact on green innovation performance.

2.3 EAT and EC

Environmental commitment has gradually become one of the important subjects in environmental management (1). Studies show that higher environmental education can stimulate citizenry environmental awareness (30) and promote environmental protection behavior (31, 32). 

Drawing on the KBV, EAT can increase awareness and change in behavior, thus promoting EC (33). For instance, environmental education organized by the government and society can make managers realize the importance of environmental protection, thus boosting EC. Besides, environmental protection education provided by the government and society can strengthen the managers' sense of social responsibility, which, in turn, promotes EC. Previous studies have proved that regular knowledge acquisition activities and initiatives aimed at environmental issues positively influence environmental commitment (34).

Building on the KBV, we contend that the environmental awareness training process of the Chinese medical equipment manufacturing sector can boost environmental commitment. Hence, we postulate that EAT can affect EC in an organization. We suggest the following assumptions: 

H2: Environmental awareness training has a positive impact on environmental commitment

2.3 EC and GIP

Linked to the above Hypothesis 2, drawing upon the KBV, environmental commitment is one of the most significant motives for individual intent to acquire knowledge about ecological deeds. EC refers to people’s willingness to engage in environmental protection efforts. According to the KBV, EC can influence the enterprise's environmental management and green practices, which, in turn, leads to enhanced green innovation performance. It manifests in two ways: First, it can cultivate a sense of sustainable development within enterprises. This helps enterprises to create sustainable practices and invest necessary funds in developing green products, which ultimately promotes GIP (35, 36). Second, consumers not only have a growing concern for the environment but also exhibit an increasing interest in purchasing green products (37). EC can, thus, improve the organizational image of enterprises, competitiveness, and market share, which, ultimately promotes GIP. For instance, the findings of studies conducted by Mo, Boadu (1) and Lin and Ho (38) on managerial environmental concerns and green innovation performance and green innovation reveal significant impacts among the variables. 

Drawing on the KBV, we contend that EC can aid enterprises in formulating wide-ranging ecological protection strategies in the direction of environmental matters, which, in turn, enhances enterprises' GIP. We therefore propose the following assumptions based on these assertions:

H3: Environmental commitment positively affects green innovation performance

2.3 Mediating Role of EC in EAT and GIP

Studies have indicated that EAT is a continuous education that influences managers' environmental awareness and decision-making, ultimately shaping their environmental attitudes (3, 39). Concerning external knowledge acquisition, EAT can help managers acquire knowledge related to environmental protection concepts and practices. Therefore, EAT can change their behavioral awareness (32), which affects their values and environmental attitudes (40), thus promoting EC.

EAT further establishes the value orientation of green development through EC, which enhances green product development of enterprises (41, 42) and ultimately promotes GIP. The current scholarship contends that EC can positively influence enterprises' green innovation activities. Thus, enterprises can adopt some responsible production techniques to enhance GIP. Extant works have proved how managerial environmental concerns affect the enterprise's GIP. For instance, Xie, Chen (4) and Mo, Boadu (1) find a snooping consequence of managerial environmental concerns in the correlation between pro-environmental education and eco-friendly agricultural production and CSR activities and GIP, respectively. 

Drawing from the internal knowledge transfer, we contend that EAT can promote EC while in the internal knowledge sharing and exchange, it makes environmental protection the consensus of managers to establish the image of enterprises in the hearts of the people and the government, thus promoting GIP. We therefore suggest the hypothesis below: 

H4: Environmental commitment mediates the relationship between environmental awareness training and green innovation performance

2.3 Moderating role of EKEA

EAT ensures the transfer of knowledge to enterprise managers through training. Training creates environmental protection awareness, which enhances the sense of social responsibility in enterprise managers. The enhanced social responsibility of enterprise managers improves their inclination to formulate ecological strategies, which ultimately promotes EC (43). However, most knowledge transmitted to managers through education and training is explicit. Explicit knowledge is open and lacks creativity (44), which is not conducive to transmission and reception. On the contrary, tacit knowledge is the source of several opportunities and potentials found and created. Therefore, in knowledge management, it is a great challenge to make explicit knowledge implicit so that knowledge can be effectively received and used in the transmission process (45). This paper introduces the variable EKEA, which can effectively solve this obstacle.

With reference to Buder (46), this paper defines EKEA as the behavior of changing one's environmental awareness or interacting with others. EKEA is a dynamic and flowing process that combines different forms of knowledge from multiple sources to effectively improve the efficiency and practicality of knowledge transfer (47). Existing research has shown that EKEA is critical in extending the knowledge spillover effect within organizations and building competitive advantages (48). Additionally, EKEA has the potential to facilitate collaboration between individuals with diverse backgrounds and skills, which results in the creation of new knowledge and more efficient sharing and utilization of existing knowledge. This enhances the effectiveness of knowledge transfer and acquisition (49). Therefore, when EKEA is limited, the knowledge acquired by managers is mainly reflected at the explicit level, which results in low efficiency of knowledge transmission and acceptance. This adversely affects the impact of EAT on EC. Conversely, when there is an abundance of EKEA, it enhances knowledge transfer efficiency through the transfer mode of mutual exchange and sharing of implicit knowledge. This enhances the impact of EAT on EC. Therefore, we suggest the assumptions below: 

H5: EKEA plays a positive regulatory role in the link between EAT and EC, such that strengthening EKEA enhances, the positive effect of EAT on EC, while weakening EKEA decreases this impact. 

3.2 Participants and Procedure

This research has received approval from the Hospital Academic Ethics Committee. In testing the hypothesis, this study uses China's medical equipment enterprises as a survey object through quantitative research methods. In ensuring the clarity of survey questions, focus group discussions with five medical equipment suppliers were undertaken. After several rounds of revisions, a pilot survey was conducted. The pilot survey entailed interviewing 42 senior executives of medical equipment suppliers from two top three hospitals in southwest China. The data analysis results preliminarily verified the study hypothesis. 

Furthermore, we obtained a directory of senior executives of medical equipment suppliers from the Alumni platform of several double first-class universities in the southwest region. Through random sampling, we selected 400 senior executives from the directory and invited them to participate in the survey via email or phone. After receiving a positive response via email or phone, 377 questionnaires were distributed to senior executives of medical equipment enterprises from March 2023 to April 2023. Finally, we received a total of 296 questionnaires from the senior executives. After deleting invalid questionnaires, the study retained 285 valid samples, representing a participation rate of 75.60%. The characteristics of the respondents are presented in Table 1.

4. Results 

- Large sample size: This paper collected data from only 285 medical equipment manufacturing companies in China. Are results presented clearly and analyzed appropriately? Do the conclusions adequately tie together the other elements of the paper? 

- Are results presented clearly and analyzed appropriately? Do the conclusions adequately tie together the other elements of the paper?: The results are presented clearly and analyzed appropriately. The conclusions adequately tie together the other elements of the paper and are supported by the results and analysis presented. The authors have effectively interpreted the results just need to provided a clear and concise summary of the findings. Overall, the results and conclusions are well-supported and contribute to the overall strength of the paper.

Respond: Thanks for your commendation. 

5. Implications for research 

- Does the paper identify clearly any implications for research? Does the paper bridge the gap between theory and practice? How can the research be used in practice (economic and commercial impact), in teaching, to influence public policy, in research (contributing to the body of knowledge)? 

- Are these implications consistent with the findings and conclusions of the paper?: The paper clearly identifies implications for research. The authors was unable to effectively bridged the gap between theory and practice by providing practical applications for the research findings. Moreover, the discussion could benefit from a more in-depth reflection on the theoretical implications of the study. For instance, how do the findings contribute to the existing literature on the use of environmental awareness training and environmental commitment on firm’s green innovation performance in the context of medical devices suppliers? What are the theoretical or practical implications of the newly established relationships in the KBV model?

Respond: Thank you very much for your comment and suggestion. We have enhanced and discussed the theoretical or practical implications of the study in detail. It reads:

5.2 Theoretical implications

Second, we have contributed to research related to EAT and GIP. Our empirical analysis reveals that EAT has a positive influence on GIP. This conclusion is somewhat in line with prior academic works such as Xie, Chen (4) and (3), which highlight the pivotal role that pro-environmental education plays in influencing environmental protection activities. Our study adds nuance to the extant works in several key ways. Foremost among these is the targeted focus on the impact of EAT on GIP, which has hitherto been underexplored, particularly within the medical equipment manufacturing sector in emerging economies. Our results extend KBV by demonstrating that it plays a key function in the impact of EAT on GIP. 

Third, this research depicts the internal influence mechanism of EAT on GIP from the perspective of KBV. Existing research has conducted extensive research on GIP based on methods (56, 57), knowledge management (58), quality management (59), green core competence (29), and other aspects. The current research neglects the influence of the managerial subjective perception. The environmental education and training for enterprise managers will inevitably affect their objective conscious behavior and values. Research on internal influence mechanisms is needed to attain a holistic understanding of EAT. This paper reveals the internal process mechanism of "Environmental awareness training—Environmental commitment—Green innovation performance" based on the KBV perspective, filling this research gap, which is somewhat in line with prior academic works such as Xie, Chen (4) and Mo, Boadu (1) that highlight an interfering consequence of managerial environmental concerns in the correlation between pro-environmental education and eco-friendly agricultural production and CSR activities and GIP, respectively. Our study delves deeper by identifying that EC mediates the correlation between EAT and GIP. This indicates that EAT and EC are key stepping stones for the GIP to take place effectively. The study is an add-on to KBV and related studies on GIP.

Finally, utilizing the knowledge-based view as a theoretical framework, this research analyzed the moderating effect of EKEA and identified a boundary condition that enhances the impact of EAT on EC. EAT by government organizations promotes the acceptance of relevant environmental protection concepts, thus effectively promoting EC. However, in this process, most of the knowledge is explicit knowledge, ignoring the implicit process of explicit knowledge (60), which ultimately leads to problems such as low efficiency and low availability of EAT knowledge in the transmission process and acceptance. Based on this, we introduced the variable EKEA, which can help enterprise executives solve challenges that arise during the knowledge transfer process (61). From the knowledge-based perspective, this paper examines the reinforcement of EKEA on EAT and EC and the mediation effect of EKEA on “Environmental awareness training—Environmental commitment—Green innovation performance.” These findings contribute to extending the scope of the KBV in environmental education. 

4.3 Management implications

There are three main management implications of this study:

Firstly, the research findings indicate that EAT is significantly associated with GIP. We suggest civil society organizations and government agencies organize environmental protection awareness, sensitization, and training programs for enterprise managers to grip EAT concepts in their green innovation policies. 

Secondly, the results of this study show that EKEA moderates the relationship between EAT and EC and mediates the relationship between EAT and GIP, thus playing a significant role in regulating the overall process of "Environmental awareness training—Environmental commitment — Green innovation performance.” This shows that environmental knowledge exchange activities are conducive to managers' assimilation, accelerate the knowledge-gathering process, and thus improve the efficiency of knowledge transmission and reception. We suggest that enterprises consider EKEA as a process in the publicity activities of environmental protection knowledge. The exchange of training experiences among participants after the training sessions can aid in improving managers’ green innovation awareness, thus promoting EC.

Finally, authorities or government departments should formulate relevant incentive measures, implement the green development policy, and integrate environmental protection awareness and sensitization into their strategic framework. And apply it to develop the capacity of enterprise managers, which, in the long run, improves the GIP of businesses.

6. Discussion of Findings

- Moreover, the discussion could benefit from a more in-depth reflection on the theoretical implications of the study. For instance, how do the findings contribute to the existing literature on the use of the use of environmental awareness training and environmental commitment on firm’s green innovation performance in the context of medical devices suppliers? What are the theoretical or practical implications of the newly established relationships in the KBV model?

- The study in this paper is limited to medical equipment manufacturing companies in China, so the findings may not be applicable to other user groups. 

In summary, although this paper provides valuable insights in examining the effect of environmental awareness training and environmental commitment on firm’s green innovation performance,more work is needed to expand on the literature review, improve the paper structure and writing, and deepen the theoretical contributions.

Respond: Thank you very much for your comment and suggestion. We have enhanced and discussed the theoretical or practical implications of the study in detail. Besides, we have identified the paper's focus on medical equipment manufacturing companies in China as a limitation. It reads:

5.2 Theoretical implications

Second, we have contributed to research related to EAT and GIP. Our empirical analysis reveals that EAT has a positive influence on GIP. This conclusion is somewhat in line with prior academic works such as Xie, Chen (4) and (3), which highlight the pivotal role that pro-environmental education plays in influencing environmental protection activities. Our study adds nuance to the extant works in several key ways. Foremost among these is the targeted focus on the impact of EAT on GIP, which has hitherto been underexplored, particularly within the medical equipment manufacturing sector in emerging economies. Our results extend KBV by demonstrating that it plays a key function in the impact of EAT on GIP. 

Third, this research depicts the internal influence mechanism of EAT on GIP from the perspective of KBV. Existing research has conducted extensive research on GIP based on methods (56, 57), knowledge management (58), quality management (59), green core competence (29), and other aspects. The current research neglects the influence of the managerial subjective perception. The environmental education and training for enterprise managers will inevitably affect their objective conscious behavior and values. Research on internal influence mechanisms is needed to attain a holistic understanding of EAT. This paper reveals the internal process mechanism of "Environmental awareness training—Environmental commitment—Green innovation performance" based on the KBV perspective, filling this research gap, which is somewhat in line with prior academic works such as Xie, Chen (4) and Mo, Boadu (1) that highlight an interfering consequence of managerial environmental concerns in the correlation between pro-environmental education and eco-friendly agricultural production and CSR activities and GIP, respectively. Our study delves deeper by identifying that EC mediates the correlation between EAT and GIP. This indicates that EAT and EC are key stepping stones for the GIP to take place effectively. The study is an add-on to KBV and related studies on GIP.

Finally, utilizing the knowledge-based view as a theoretical framework, this research analyzed the moderating effect of EKEA and identified a boundary condition that enhances the impact of EAT on EC. EAT by government organizations promotes the acceptance of relevant environmental protection concepts, thus effectively promoting EC. However, in this process, most of the knowledge is explicit knowledge, ignoring the implicit process of explicit knowledge (60), which ultimately leads to problems such as low efficiency and low availability of EAT knowledge in the transmission process and acceptance. Based on this, we introduced the variable EKEA, which can help enterprise executives solve challenges that arise during the knowledge transfer process (61). From the knowledge-based perspective, this paper examines the reinforcement of EKEA on EAT and EC and the mediation effect of EKEA on “Environmental awareness training—Environmental commitment—Green innovation performance.” These findings contribute to extending the scope of the KBV in environmental education. 

4.3 Management implications

There are three main management implications of this study:

Firstly, the research findings indicate that EAT is significantly associated with GIP. We suggest civil society organizations and government agencies organize environmental protection awareness, sensitization, and training programs for enterprise managers to grip EAT concepts in their green innovation policies. 

Secondly, the results of this study show that EKEA moderates the relationship between EAT and EC and mediates the relationship between EAT and GIP, thus playing a significant role in regulating the overall process of "Environmental awareness training—Environmental commitment — Green innovation performance.” This shows that environmental knowledge exchange activities are conducive to managers' assimilation, accelerate the knowledge-gathering process, and thus improve the efficiency of knowledge transmission and reception. We suggest that enterprises consider EKEA as a process in the publicity activities of environmental protection knowledge. The exchange of training experiences among participants after the training sessions can aid in improving managers’ green innovation awareness, thus promoting EC.

Finally, authorities or government departments should formulate relevant incentive measures, implement the green development policy, and integrate environmental protection awareness and sensitization into their strategic framework. And apply it to develop the capacity of enterprise managers, which, in the long run, improves the GIP of businesses.

4.4 Limitation

Finally, this study was conducted in a single geographic context (China) in a single sector (medical equipment manufacturing companies in China). Therefore, researchers must be cautious when generalizing these results and conclusions to other settings. Replicating this study in a different geographic context or sector would be helpful to generalize our insights and conclusions.

Reviewer #2: The article is interesting and describes an important research problem. However, I recommend making a few changes and additions.

Respond: Thanks for your comment and good suggestion. The current version has made significant improvements compared to before. Thank you again.

1. The abstract should not repeat what has been said in the research background. The first two paragraphs in the research background part are the same as in the abstract. So, you need to rewrite the research background part to not repeat the abstract.

Respond: Thanks for your comment and good suggestion. We have updated the Research background. It reads:

1. Research background

The significance of green innovation cannot be overemphasized in developed and emerging economies. It plays a critical function in the twenty-first century for enterprises’ success, development, and survival. Mo, Boadu (1) assert that green innovation manages the environment, energy usage, pollutant production, and waste disposal and recycling towards cleaner production and sustainability. Thus, it serves as a superior player, mitigates the production of pollutants, and increases cost efficiency and competitive advantage. For the past few years, stakeholders have gradually become aware of green innovation’s importance to society and how enterprises should operate in an environmentally friendly manner to integrate green practices into strategic and process management. However, compared with developed markets environment, emerging markets face challenges between development and the environment. For instance, with the escalation of global industrialization, Chinese enterprises' production and operating activities have caused pollution, threatening the life of the earth's ecosystem (2). Due to the alarming nature of the menace, for enterprises to survive, scholars, government agencies, and civil society organizations have recognized environmental protection sensitization and awareness training programs as a vital weapon for curbing the rising environmental issues to promote sustainability. Environmental awareness training refers to the sum of the training and learning actions on ecological issues, hands-on activities for sustainability, and relevant environmental policies and regulations taken by enterprises to meet environmental challenges. Environmental awareness training activities offer corrective measures to mitigate ecological menace towards innovation performance. Previous studies have established the positive impact of pro-environmental education on environmental protection (3, 4). However, there is a dearth of studies on the environmental awareness training effectiveness on the green innovation performance (GIP) of enterprises, especially in emerging economies. This research aims to bridge the gap by exploring the cause-and-effect relationship between environmental awareness training (EAT), environmental knowledge exchange activities (EKEA), and environmental commitment (EC) in GIP and related boundary conditions using the Chinese medical equipment manufacturing industry. Chinese market environment offers a fertile ground for the study of such nature to understand the linkage amongst the variables. 

2. Discussion should be included in the main body of the paper. More in-depth discussion should be included to support the interpretations and conclusions. You should start with an overall comment about the findings and then critically evaluate the main issues raised by the study. You should also present a coherent argument for their position.

Respond: Thanks for your comment and good suggestion. We have added some in-depth discussions in the theoretical section, including comparisons with existing research, and the added contributions of this study. It reads:

Second, we have contributed to research related to EAT and GIP. Our empirical analysis reveals that EAT has a positive influence on GIP. This conclusion is somewhat in line with prior academic works such as Xie, Chen (4) and (3), which highlight the pivotal role that pro-environmental education plays in influencing environmental protection activities. Our study adds nuance to the extant works in several key ways. Foremost among these is the targeted focus on the impact of EAT on GIP, which has hitherto been underexplored, particularly within the medical equipment manufacturing sector in emerging economies. Our results extend KBV by demonstrating that it plays a key function in the impact of EAT on GIP. 

Third, this research depicts the internal influence mechanism of EAT on GIP from the perspective of KBV. Existing research has conducted extensive research on GIP based on methods (56, 57), knowledge management (58), quality management (59), green core competence (29), and other aspects. The current research neglects the influence of the managerial subjective perception. The environmental education and training for enterprise managers will inevitably affect their objective conscious behavior and values. Research on internal influence mechanisms is needed to attain a holistic understanding of EAT. This paper reveals the internal process mechanism of "Environmental awareness training—Environmental commitment—Green innovation performance" based on the KBV perspective, filling this research gap, which is somewhat in line with prior academic works such as Xie, Chen (4) and Mo, Boadu (1) that highlight an interfering consequence of managerial environmental concerns in the correlation between pro-environmental education and eco-friendly agricultural production and CSR activities and GIP, respectively. Our study delves deeper by identifying that EC mediates the correlation between EAT and GIP. This indicates that EAT and EC are key stepping stones for the GIP to take place effectively. The study is an add-on to KBV and related studies on GIP.

Finally, utilizing the knowledge-based view as a theoretical framework, this research analyzed the moderating effect of EKEA and identified a boundary condition that enhances the impact of EAT on EC. EAT by government organizations promotes the acceptance of relevant environmental protection concepts, thus effectively promoting EC. However, in this process, most of the knowledge is explicit knowledge, ignoring the implicit process of explicit knowledge (60), which ultimately leads to problems such as low efficiency and low availability of EAT knowledge in the transmission process and acceptance. Based on this, we introduced the variable EKEA, which can help enterprise executives solve challenges that arise during the knowledge transfer process (61). From the knowledge-based perspective, this paper examines the reinforcement of EKEA on EAT and EC and the mediation effect of EKEA on “Environmental awareness training—Environmental commitment—Green innovation performance.” These findings contribute to extending the scope of the KBV in environmental education. 

3. You should clearly present all the measurement items of variables (e.g. in a table of Appendix).

Respond: Thanks for your comment and good suggestion. We have added the “Appendix: Measurement” in the attachment. It reads:

Appendix: Measurement 

Environmental awareness training

Enterprise managers actively participate in the environmental protection theoretical training organized by government agencies or other organizations.

Enterprise management personnel actively participate in pro-environmental practice which organized by the government agencies or other organizations.

Enterprise managers actively participate in the skills training in low-carbon energy conservation arranged by government institutions or non-governmental organizations.

Enterprise managers actively participate in environmental policy-related education and awareness programs, which are arranged by government institutions or non-governmental organizations.

Environmental Commitment

Enterprises pay close attention to environmental interests in production and operation.

Enterprises’ corporate culture and values deeply reflect the performance of environmental protection responsibilities.

Enterprises are willing to continuously invest manpower, material resources, and funds to fulfill environmental responsibilities.

Enterprises consider environmental protection to be a crucial aspect of its long-term strategy.

Environmental knowledge exchange activities 

Enterprises actively organize employees to share knowledge and experience beneficial to environmental protection.

Enterprises encourage employees to discuss knowledge about environmental protection measures or technologies.

Enterprises actively participate in environmental protection related exchange meetings and programs which arranged by government institutions or non-governmental organizations.

Enterprises actively participate in the sharing activities of environmental protection related knowledge and technology organized by the industry.

Green innovation performance

New products or businesses developed by the enterprises focused on the use of environmentally friendly materials.

New products or businesses developed by the enterprises attached great importance to energy conservation.

The new products or businesses developed by the enterprises reasonably dispose of hazardous substances or waste materials.

The new products or businesses developed by the enterprises attach great importance to the recyclability and reusability of the goods sold.

Thanks for your careful and professional guidance on our paper. We have made every effort to address all issues. The current version has made significant improvements compared to before. Thank you again.

---

## [Decision Letter · Decision Letter 1]

16 Jan 2024

Effects of environmental awareness training and environmental commitment on firm’s green innovation performance: Empirical insights from medical device suppliers

PONE-D-23-15337R1

Dear Dr. Chen,

We’re pleased to inform you that your manuscript has been judged scientifically suitable for publication and will be formally accepted for publication once it meets all outstanding technical requirements.

Kind regards,

Xingwei Li, Ph.D.

Academic Editor

PLOS ONE

Additional Editor Comments (optional):

Reviewers' comments:

Reviewer's Responses to Questions

**Comments to the Author**

1. If the authors have adequately addressed your comments raised in a previous round of review and you feel that this manuscript is now acceptable for publication, you may indicate that here to bypass the “Comments to the Author” section, enter your conflict of interest statement in the “Confidential to Editor” section, and submit your "Accept" recommendation.

Reviewer #1: All comments have been addressed

Reviewer #3: All comments have been addressed

2. Is the manuscript technically sound, and do the data support the conclusions?

Reviewer #1: Yes

Reviewer #3: Yes

3. Has the statistical analysis been performed appropriately and rigorously? 

Reviewer #1: Yes

Reviewer #3: Yes

4. Have the authors made all data underlying the findings in their manuscript fully available?

Reviewer #1: Yes

Reviewer #3: Yes

5. Is the manuscript presented in an intelligible fashion and written in standard English?

Reviewer #1: Yes

Reviewer #3: Yes

6. Review Comments to the Author

Reviewer #1: Authors have statistical analysis have been performed appropriately and rigorously, the manuscript is presented in an intelligible fashion and written in standard English Language proficiency

Reviewer #3: The topic is very interesting, and I like the topic and appreciate your efforts to present your revised research work nicely. The overall work is good. The authors have clarified their interpretations and applications by clarifying the limitations of certain methods, adding a helpful conceptual model, and improving the discussion.

7. PLOS authors have the option to publish the peer review history of their article (what does this mean?). If published, this will include your full peer review and any attached files.

Reviewer #1: No

Reviewer #3: No

---

## [Editor Report · Acceptance letter]

22 Feb 2024

PONE-D-23-15337R1 

PLOS ONE

Dear Dr. Chen, 

I'm pleased to inform you that your manuscript has been deemed suitable for publication in PLOS ONE. Congratulations! Your manuscript is now being handed over to our production team.

Kind regards, 

on behalf of

Prof. Dr. Xingwei Li 

Academic Editor

PLOS ONE